# Designing Optimal Breakfast for the United States Using Linear Programming and the NHANES 2011–2014 Database: A Study from the International Breakfast Research Initiative (IBRI)

**DOI:** 10.3390/nu11061374

**Published:** 2019-06-19

**Authors:** Florent Vieux, Matthieu Maillot, Colin D. Rehm, Adam Drewnowski

**Affiliations:** 1MS-Nutrition, 27 bld Jean Moulin Faculté de Médecine la Timone, Laboratoire C2VN, 13385 Marseille CEDEX 5, France; florent.vieux@ms-nutrition.com (F.V.); matthieu.maillot@ms-nutrition.com (M.M.); 2Albert Einstein College of Medicine, Montefiore Medical Center, New York, NY 10467, USA; colin.rehm@gmail.com; 3Center for Public Health Nutrition, University of Washington, Box 353410, Seattle, WA 98195, USA

**Keywords:** breakfast, linear programming, NHANES, NRF9.3, nutrient density, food groups, nutrients, optimization

## Abstract

The quality of dietary patterns can be optimized using a mathematical technique known as linear programming (LP). LP methods have rarely been applied to individual meals. The present LP models optimized the breakfast meal for those participants in the nationally representative National Health and Nutrition Examination Survey 2011–2014 who ate breakfast (*n* = 11,565). The Nutrient Rich Food Index (NRF9.3) was a measure of diet quality. Breakfasts in the bottom tertile of NRF9.3 scores (T1) were LP-modeled to meet nutrient requirements without deviating too much from current eating habits. Separate LP models were run for children and for adults. The LP-modeled breakfasts resembled the existing ones in the top tertile of NRF9.3 scores (T3), but were more nutrient-rich. Favoring fruit, cereals, and dairy, the LP-modeled breakfasts had less meat, added sugars and fats, but more whole fruit and 100% juices, more whole grains, and more milk and yogurt. LP modeling methods can build on existing dietary patterns to construct food-based dietary guidelines and identify individual meals and/or snacks that need improvement.

## 1. Introduction

Breakfast consumers in the US and globally exhibit a variety of eating patterns [1,2]. Analyses of the National Health and Nutrition Surveys (NHANES) in the US suggest that those patterns typically include grain products, consumed either alone, or with fruit juice, milk, whole fruit, sweets, meat and eggs, and coffee or tea [3,4]. Given that breakfast continues to be thought of as the most important meal of the day [5,6], identifying optimal food patterns at breakfast continues to be a topic of research interest [1,2].

The International Breakfast Research Initiative (IBRI) recently examined the food and nutrient composition of breakfasts eaten in Canada [7], Denmark [8], France [9], Spain [10], the United Kingdom [11], and the United States [12]. Nationally representative dietary intake databases were used. Breakfasts associated with highest-quality diets were characterized as to their food and nutrient content. The summary paper made recommendations for a “global” healthy breakfast, based on multi-country findings [2,13]. Those were empirical dietary recommendations based on observed dietary intakes for each population.

The quality of daily diets can also be optimized using a mathematical technique known as linear programming (LP) [14,15,16]. LP methods strive to find the optimal combination of daily foods for a given population subject to a variety of constraints [17]. For example, the US Department of Agriculture’s Thrifty Food Plan (TFP), a variant of an LP model, was developed to identify the lowest-cost nutritionally adequate diet, while respecting existing eating habits [18,19]. Given adequate dietary data, nutritionally optimal diets can also be constructed for populations, population subgroups, or even for individual respondents [15]. Typically, the optimized diets need to meet energy and nutrient requirements at low cost, while minimizing deviation from existing diets [16]. 

Thus far, LP models have been applied to dietary patterns at the population or at the individual level [14,20]. There are few examples where LP methods were applied to individual meals. In a novel application, we used LP to optimize breakfast meals associated with low-quality diets in the 2011–2014 NHANES database. The question was whether the LP-modeled breakfasts would resemble existing ones in the top tertile (T3) of diet quality, or would they follow an altogether different path? In general, dietary guidance that is based on existing eating habits is more feasible and easier to implement than is dietary guidance that breaks entirely with habit, tradition, and culture [21,22].

## 2. Materials and Methods 

### 2.1. Study Population and Dietary Data 

Analyses were based on the first day of dietary intakes in the 2011–2012 and 2013–2014 cycles of the nationally representative National Health and Nutrition Examination Survey (NHANES) [23,24]. The first 24-h recall in the NHANES was completed in-person at the Mobile Examination Center with a trained interview. The 24-h recall queries all foods/beverages consumed by participants from midnight-to-midnight on the previous day [25,26]. Dietary supplements were excluded. Breakfast was defined as the self-reported “breakfast/desayuno” and brunch. An energy threshold of 50 kcal was imposed. Breakfast skippers were defined as having no breakfast or an eating episode of <50 kcal.

Data were available for 14,488 children, adolescents, and adults aged ≥6 years. The sample included 4057 children (ages 6–17 years) and 10,431 adults (ages >18 years). Of those, 11,565 persons were previously identified as breakfast consumers. The present analytical sample was therefore based on 3296 children and 8269 adults.

The population sample was stratified by 2 age groups (6–17 years and >18 years) and six race/ethnicity groups (non-Hispanic white, non-Hispanic black, Mexican-American, other Hispanic, Asian, and other/mixed race). Education was defined as: <High School (<12 years), High School (12 years); Some college (12–16 years), and >College (>16 years). Income-to-poverty ratio (IPR) cut-points were set at: <1.3; 1.3–1.849; 1.85–2.99; and >3.

### 2.2. Food Categories and Food Groups

Food categories and food groups were derived from the What We Eat in America (WWEIA) food items after exclusion of “alcoholic beverages”, “baby beverages”, “no category”, “other”, “baby food”, “baby beverages”, “infant formula”, “condiments and sauces”, and “water” (the number of food categories used in the analysis is 31) [27]. The US Department of Agriculture (USDA) has different ways of assigning foods into aggregate categories. We chose to use the What We Eat in America (WWEIA) scheme, since it was more granular than the USDA MyPlate scheme and therefore better suited to linear programming [28]. Whereas the MyPlate scheme does split grains into refined and whole grains, the present WWEIA scheme does not [29]. By contrast, the present scheme separates dairy and protein foods into multiple categories, whereas MyPlate does not. In the present LP model, the frequency weighted “milk” contained about 1 g of Saturated Fatty Acids (SFA) and 48 kcal for 100 g. In order to reduce SFA intake, milk effectively replaced cheese (which contains about 12 g of SFA and 290 kcal per 100 g and which was removed in absolute function models), as well as processed meat (7 g of SFA and 290 kcal per 100 g and which was removed in all models). Milk also replaced sweet bakery (7 g of SFA and 400 kcal per 100 g), eggs (4.5 g of SFA and 180 kcal per 100 g), and mixed dishes (3.7 g of SFA and about 215 kcal per 100 g).

### 2.3. Measures of Dietary Quality

Energy and nutrient intakes for NHANES participants were calculated using the Food and Nutrient Database for Dietary Studies (FNDDS) 2011–2014, customized with the addition of vitamin D and added sugar data [30]. This information was supplemented with data from the Food Patterns Equivalents Database (FPED) from the USDA [31].

The Nutrient Rich Foods (NRF9.3) index, was the principal measure of diet quality [1,32]. The NRF9.3 is based on 9 qualifying and 3 disqualifying nutrients. Reference daily values (DVs) were based on the US Food and Drug Administration (FDA) and other standards [12,32]. The qualifying nutrients and standard reference amounts were as follows: Protein (50 g), fiber (28 g), vitamin A (900 RAE), vitamin C (90 mg), vitamin D (20 mcg), calcium (1300 mg), iron (18 mg), potassium (4700 mg), and magnesium (420 mg). The 3 disqualifying nutrients and maximum recommended values (MRVs) were: Added sugar (50 g), saturated fat (20 g), and sodium (2300 mg). The NRF9.3 was calculated as follows: 

with
NRF 9.3 = (NR − LIM) × 100(1)
(2)$$NR=\sum_{i=1}^{9}\frac{Intake_{i}/Energy\times 2000}{DV_{i}}$$and(3)$$LIM=\overset{3}{\underset{i=1}{\sum }}\frac{Intake_{i}/Energy\times 2000}{MRV_{i}}-1$$
where intake_i_ is the daily intake of each nutrient i, and DVi is the reference daily value for that nutrient. In the nutrients-to-encourage (NR) calculation, each daily nutrient intake i was adjusted for 2000 kcal and expressed as a percentage of the DV. Following past protocol, percent DVs for nutrients were truncated at 100, so that an excessively high intake of one nutrient could not compensate for the dietary inadequacy of another. In the nutrients-to-limit (LIM) calculation, only the share in excess of the recommended amount was considered

In the present adaptation, vitamin D, a nutrient of public health concern [33,34], replaced vitamin E. Fiber, vitamin D, calcium, magnesium, and potassium were all identified in the 2010 Dietary Guidelines for Americans (DGA) as nutrients of concern [33]. The NRF score was adjusted for energy intakes, analogous with the recent versions of the USDA Healthy Eating Index (HEI) [35]. Age-specific tertiles of NRF9.3 served to stratify children and adults by overall diet quality (T1, T2, and T3). 

### 2.4. Linear Programming Applied to T1 Breakfast

Separate LP analyses were run for children and adults. The LP model was used to derive optimized breakfasts for children and adults in the bottom tertile (T1) of diet quality, as indexed by NRF9.3 scores. Table 3 shows that the LP-modeled breakfasts met nutrient recommendations established by the IBRI group. The %DVs were taken from “Food Labeling: Revision of the Nutrition and Supplement Facts Labels” [36]. For nutrients expressed in percentage of energy, the recommendations derived by the IBRI were used.

To ensure that the LP-optimized breakfasts remained as close as possible to the observed breakfast food patterns, two mathematical functions were applied. The more often used relative function favors the selection by the LP model of foods that are already eaten in reasonable quantities. In other words, the relative function avoids incorporating in the LP model those foods that are eaten rarely or not at all.

“Absolute function”:(4)$$\min\;D=\overset{31}{\underset{i=1}{\sum }}abs(optimized\;quantity^{i}-Observed\;quantity^{i})$$

“Relative function”:(5)$$Min\;D=\overset{31}{\underset{i=1}{\sum }}\frac{abs(optimized\;quantity^{i}-observed\;quantity^{i})}{observed\;quantity^{i}}$$
where each individual food item and D is the distance.

Compared to the absolute function, the relative function is more likely to modify those foods that are already consumed in large quantities (or to excess). For example, an individual can obtain 480 mg of calcium from one serving of milk (250 g) and one serving of cheese (30 g). In order to obtain 510 mg calcium (breakfast target), the relative function will increase the amount of milk to 275 g (+10%) and not change the amount of cheese (0.1 is a smaller value than increasing cheese by 17%). The absolute one will increase the quantity of cheese by 5 g (5 g is a smaller value than 25 g of milk). Optimized breakfasts were derived by using those two functions. 

### 2.5. Analytical Approach

All analyses were conducted using SAS software, version 9.4, and are representative of the US population (SAS Institute Inc., Cary, NC, USA). Differences in NRF scores between socio-demographics groups were tested using linear regression.

### 2.6. Data Availability and Ethical Approval

The necessary Institutional Review Board (IRB) approval for NHANES had been obtained by the National Center for Health Statistics (NCHS) [24]. For adult participants, written informed consent was obtained directly from the participating adult. For child participants, parental/guardian written informed consent was obtained and children/adolescents ≥12 years provided additional written consent. All data used here are publicly available on the NCHS and USDA websites [23,37]. Publicly available data, such as those used here per University of Washington policies, do not involve “human subjects” and their use requires neither IRB review nor an exempt determination. According to University of Washington policies, these data may be used without any involvement of the Human Subjects Division or the University of Washington Institutional Review Board.

## 3. Results

Table 1 shows mean NRF9.3 scores for total diets of breakfast consumers by gender, age group, and socio-demographics. Gender effects depended on age, where adult women had more nutrient-dense diets than did men, whereas no gender differences were observed for children (<18 years old). The most nutrient-dense diets were consumed by Asians. Non-Hispanic blacks had the lowest quality diets at every age. Diet quality of adults greatly improved with education and with household incomes. An income gradient for children was not observed. For adults, differences in NRF scores by education and incomes were greater than those observed by race/ethnicity.

Also shown are NRF scores for breakfast consumers in the bottom tertile (T1).

### Comparing Existing and LP-Modeled Breakfasts

Figure 1 shows differences in the composition of breakfasts in the bottom (T1), middle (T2), and the top (T3) tertile of NRF9.3 scores. The data are shown separately for breakfast-consuming children and adults. There were a number of progressive changes in breakfast composition on going from T1 to T3 of diet quality. First, the consumption of milk and yogurt increased, and cheese dropped slightly. Meat and eggs were sharply reduced. The consumption of soy, nuts, and seeds was substantially higher for adults. Refined grains showed a very sharp drop, whereas the amounts of whole grains doubled and tripled. The breakfast consumption of citrus fruit, fruit juice, and other fruits was sharply increased. 

Figure 2 shows the composition of existing T1 breakfasts and linear programming (LP) modeled breakfasts, separately for children and for adults. Two models were used, a linear programming model with the relative objective function (LP-R) and a linear programming model with absolute function (LP-A). First, the modeled amounts of fluid milk were much higher than those observed, especially for the LP-R model. Yogurt was increased slightly, but cheese dropped. Meat and eggs were very sharply reduced by both models. The modeled breakfast amounts of soy, nuts and seeds were largely unchanged from T1 in model LP-R but were greatly increased in model LP-A. Refined grains showed a very sharp drop in both models, whereas the amounts of whole grains were much higher. The modeled amounts of citrus fruit and other fruits sharply increased in both models. The amounts of fruit juice were unchanged in both models.

Table 2 shows which specific breakfast foods were increased or reduced by the LP optimization model, or eliminated altogether. For children, the amounts of milk, whole fruit, and RTE (ready-to-eat) cereals were sharply increased. Sweet bakery goods, mixed dishes, processed breakfast meats and eggs dropped to zero. Quick breads were reduced. No other major changes were observed. For adults, the amounts of milk, whole fruit, and RTE cereals were sharply increased. Sweet bakery goods, processed breakfast meats, mixed dishes, quick breads, and eggs dropped to zero. No other major changes were observed.

Table 3 shows the nutrient composition of existing and LP-optimized breakfasts. For adults, the most difficult nutrient recommendation to fulfill were those for fiber, vitamin D, and sodium. As shown in Table 3, for those nutrients the LP-modeled content was strictly equal to the recommendation. For children, the limiting breakfast nutrients were fiber, potassium, magnesium, and sodium. Both models were limited by energy.

## 4. Discussion

The present analyses showed that breakfasts associated with higher-quality diets were replicated, for the most part, through LP modeling. The NRF was the measure of diet quality. Lower-quality diets were those in the bottom tertile of NRF scores (T1), whereas higher-quality diets were those in the top tertile of NRF scores (T3). Our approach was to use breakfasts associated with lower-quality diets (T1) as the point-of-departure and to compare breakfasts associated with higher-quality diets (T3) to those generated by two LP models. As expected, breakfasts associated with T3 of NRF scores were associated with higher intakes of some key nutrients than T1, including those that were in the NRF model and some that were not. The T3 breakfasts also had more food groups of interest, notably fruit, dairy, and whole grains.

Results showed that the LP models were able to improve the observed breakfast quality even more. Even though breakfast was already a relatively nutrient rich meal, the observed T3 breakfasts in children had below-recommended values for fiber, calcium, potassium, magnesium, and vitamin D, and had excess added sugars as compared to the IBRI recommendations. For adults, T3 breakfasts had too little potassium and vitamin D and too much sodium and added sugar, again as compared to the IBRI recommendations. The LP modeling showed that the observed breakfasts could be further improved, largely by changing the amounts of frequently eaten breakfast foods. 

In past studies, LP models have been applied to the optimization of daily food patterns, subject to a variety of nutritional, economic, and social constraints. LP models have also been used on the supply side, e.g., to optimize nutrient density of a school’s entire food supply [38]. The present innovation was to apply LP modeling to optimize the quality of a single meal, breakfast, as opposed to optimizing the quality of the total diet. Our use of two LP objective functions was meant to show that healthy breakfasts could be arrived at in multiple ways and following different food choices. The second innovation was to base LP modeling on those breakfast foods that were already being consumed by children and adults in the NHANES database.

For children, the typical observed breakfast foods were milk, baked goods, and sweets, with whole grain RTE cereals and whole fruit further down on the list. Adult breakfast foods included coffee/tea, sweets, fats, and white bread. Some of these observed changes were accurately tracked by the LP optimization model; others were not. First, the modeled patterns contained much greater amounts of citrus and other fruits (a several-fold increase) and same amounts of 100% juice. Breakfast whole grains almost doubled whereas refined grains dropped by half. Meat, poultry, and fish were substantially reduced, as were eggs. Soy, nuts, and legumes showed substantial increases. Milk tripled, yogurt was held constant, and the amount of cheese was reduced. The observed and the modeled patterns stressed fruit, milk, and whole grains.

Breakfast patterns created by the LP models were characterized by higher intakes of citrus fruit, whole fruit, soy, nuts, and legumes. Among children, the LP-generated breakfast patterns were characterized by higher intakes of whole grain cereals, more milk and yogurt, and lower intakes of animal protein, e.g., less meat, eggs. 

LP-A models in children increased the amounts of nuts and seeds strongly because of their high content in potassium, magnesium, and fiber, which are nutrients that are far from being reached in the observed T1 breakfast. The LP-R result display another way to reach nutrient recommendations by prioritizing foods already consumed. By providing results using different mathematical functions, our study shows that the nutritional quality of breakfast can be improved in different ways.

LP modeling of single meals is an innovation. It has often been a concern that dietary recommendations, issued by expert bodies or national governments, are hard to follow and may not actually be feasible from nutritional, behavioral, or economic standpoints. First, there were cases where multiple nutrient requirements could not be satisfied simultaneously [15]. Second, healthy foods were sometimes proposed in unrealistic large amounts [39]. Another concern was that the recommended food patterns were simply too expensive [39]. One advantage of LP models is their ability to reconcile these multiple demands without departing too far from existing eating habits. In the present study, we were able to show that the IBRI breakfast recommendations were feasible in the sense that a mathematical solution was available. In other words, we validated a set of nutrient-based recommendations by selecting the right combination of breakfast foods, with emphasis on those foods that were already consumed by children and adults. Food-based advice on healthy breakfasts can have practical implications. Healthy breakfast foods can be the starting point of public health politics. 

The present food-based approach aligns with the current dietary recommendations and guidelines which have become more food- as opposed to nutrient-based. The current research emphasis is likewise on food patterns as opposed to individual nutrients. Even though nutrient profiling (NP) models continue to be wholly nutrient based, the case can be made for advancing a hybrid NP approach that takes both nutrients and desirable food groups into account. Hybrid nutrient density scores will provide for a better alignment between NP models and the DGA, a chief instrument of food and nutrition policy in the United States. Such synergy may lead ultimately to improved dietary guidance, sound nutrition policy, and better public health.

The constraints and limitations of this study are worth noting. First, data analyses were based on the first day of the 2-day NHANES survey. Past studies on the quality of breakfast in France [9] were based on national dietary surveys that used 7-day diaries, whereas analyses of breakfast quality in the United Kingdom [11] were based on four days of food records. Second, in the present analyses we elected to go with breakfast as defined by self-report; in other studies we separated eating occasions by time of day. Third, there are different ways of aggregating foods consumed by NHANES participants into groups or categories of interest. The present input to LP models was based on What We Eat in America (WWEIA) food groups of interest, and followed the US Department of Agriculture WWEIA coding scheme [28]. The WWEIA coding scheme categorizes RTE cereals as low- versus high-sugar, whereas the narrower USDA MyPlate categories separate grains into refined and whole grains (Appendix A). Fourth, no statistical analysis comparing observed and optimized diet was possible because LP was applied to average observed diet. Developing the approach by applying LP to each individual would allow one to analyze food quantities by individual and run statistical analysis making results more robust. 

The present findings have implications for food- and meal-based dietary guidance. First, dietary guidelines are becoming more food-based, shifting emphasis from nutrients to individual meals and to composite food patterns. Our LP modeling can lead to more granular dietary advice that is provided at the level of a single meal. Furthermore, that advice can emphasize the nutritional value of foods that are already eaten, but ought to be consumed in larger or smaller amounts. Arguably, the present food-based results provide a clear indication of which foods belong in a healthy breakfast and which ought to be consumed in larger amounts or, in some cases, dropped altogether. Dietary interventions are easier when they build on existing dietary patterns and eating habits.

## 5. Conclusions

The present LP analyses showed that the IBRI recommendations for a nutritionally adequate breakfast can be met using existing breakfast foods. LP modeling can build on existing eating patterns to identify areas for potential intervention.

## Figures and Tables

**Figure 1 nutrients-11-01374-f001:**
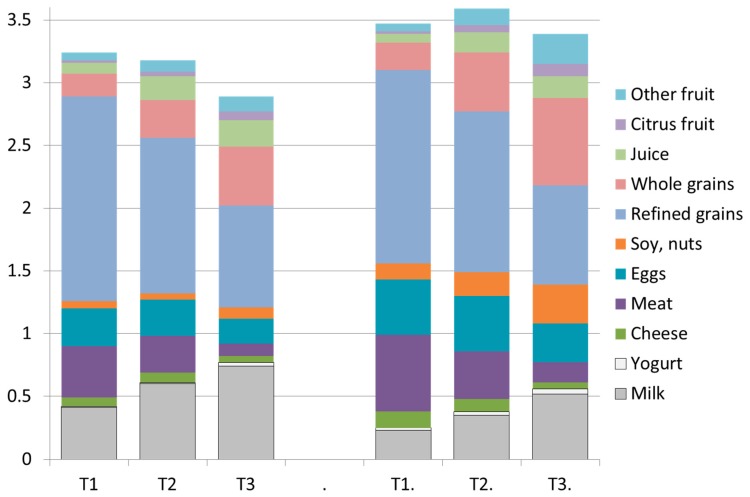
Composition of breakfasts associated with the bottom (T1), middle (T2), and top (T3) tertiles of dietary nutrient density NRF9.3 scores. Data are presented separately for breakfast consuming children (left) and for adults (right) for NHANES 2011–2014. The Y axis shows MyPlate units (cup equiv and/or ounce equiv).

**Figure 2 nutrients-11-01374-f002:**
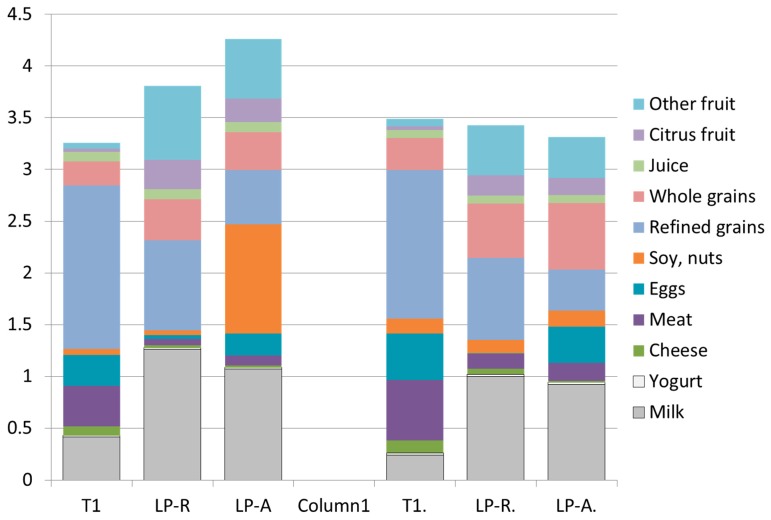
A comparison of breakfasts associated with the bottom (T1) tertile of dietary nutrient density NRF9.3 scores and breakfast patterns created by LP models. Data are presented separately for children (left) and for adults (right) in NHANES 2011–2014. The Y axis shows MyPlate units (cup equiv and/or oz equiv).

**Table 1 nutrients-11-01374-t001:** Mean (standard error) dietary nutrient density NRF 9.3 scores for breakfast consumers by age and socio-demographics. NHANES 2011–2014, United States. (NH non-Hispanic).

	All Breakfast Consumers			T1 Breakfast Consumers		
	All (11,565)	Children (3296)	Adults (8269)	T1 *n* = 4020	Children (1144)	Adults (2876)
Overall		433.34 (4.90)	444.00 (4.72)		257.80 (5.34)	254.75 (2.01)
**Gender**						
Male	5663	437.14 (5.18)	426.02 (4.99)	2,084	264.14 (8.09)	253.12 (3.23)
Female	5902	429.33 (7.06)	460.47 (5.6)	1936	251.40 (7.27)	256.69 (3.33)
		0.3057	<0.0001		0.2642	0.4977
**Race/ethnicity**						
NH White	4346	419.04 (9.94)	448.84 (6.32)	1586	248.92 (7.84)	254.36 (2.71)
NH Black	2664	413.2 (6.45)	390.32 (5.94)	1161	268.86 (7.29)	248.87 (5.01)
Mex-American	1647	475.2 (7.66)	444.35 (7.13)	472	273.36 (5.90)	258.15 (4.58)
Asian	1303	487.78 (15.42)	494.84 (6.09)	286	266.35 (12.93)	280.37 (7.15)
Hispanic	1164	449.84 (14.98)	450.69 (6.59)	357	263.15 (9.90)	268.69 (8.41)
Other	441	431.37 (17.84)	418.97 (20.58)	158	283.22 (13.18)	223.80 (21.72)
		<0.0001	<0.0001		0.0395	0.0051
**Family IPR ^1^**						
<1.3	3912	433.43 (7.32)	403.39 (6.32)	1,558	259.31 (8.86)	225.59 (5.37)
1.3–1.849	1310	440.97 (14.37)	426.8 (8.91)	478	255.79 (12.93)	253.34 (7.99)
1.85–2.99	1683	410.48 (12.55)	430.64 (8.13)	607	261.32 (10.89)	253.09 (6.34)
≥3.0	3835	439.37 (10.33)	471.28 (5.74)	1124	255.90 (6.52)	277.76 (3.79)
		0.1950	<0.0001		0.9600	<0.0001
**Education ^2^**						
<HS	1625		414.25 (5.45)	596		231.35 (5.80)
High school	1707		404.21 (7.99)	757		241.63 (4.35)
Some college	2362		436.24 (6.83)	864		256.19 (3.88)
≥College	2181		495.28 (6.93)	476		291.48 (4.73)
			<0.0001			<0.0001

^1^ IPR stands for income to poverty ratio ^2^ Missing values were removed from the analysis.

**Table 2 nutrients-11-01374-t002:** Comparisons in food composition of T1 breakfasts and the two LP models. Consumption measured in g/day. Data are presented separately for children and adults. NHANES 2011–14 United States.

What We Eat in America	Category	Children			Adults		
		T1	Optimized		T1	Optimized	
			Relative	Absolute		Relative	Absolute
Beverages	Coffee & Tea	24.3	24.3	24.3	231.4	231.4	231.4
	Diet Beverages	2.6	2.6	2.6	12.4	12.4	12.4
	Sweetened Beverages	55.9	55.9	55.9	75.5	75.5	75.5
Fats & Oils	Fats & Oils	1.7	1.7	0	9.1	9.1	9.1
Fruit	Fruit	8.9	145.7	116.2	9.3	92.1	77.1
	100% Juice	19.5	19.5	19.5	15.8	15.8	15.8
Grains	Breads	7.3	7.3	7.3	16.2	16.2	0
	Cooked grains	6.9	6.9	6.9	8.4	8.4	8.4
	Grains	0.9	0.9	0.9	1.3	1.3	1.3
	Quick Breads	20.0	13.7	0	11.6	0	0
	High Sugar RTE Cereal	6.4	24.2	6.4	3.3	22.9	3.3
	Low Sugar RTE Cereal	0.8	0.8	11.2	1.7	1.7	27.9
Milk & Dairy	Cheese	0.8	0.8	0	2.2	2.2	0
	Flavored Milk	9.7	9.7	9.7	4.5	4.5	4.5
	Milk	75.1	288.0	243.6	32.8	227.6	203.6
	Milk Dessert Drinks	0.8	0.8	0.8	0.6	0.6	0.6
	Yogurt	2.7	2.7	2.7	4.5	4.5	4.5
Mixed Dishes	Mixed Dishes	26.8	0	3.6	33.7	0	0
Protein Foods	Eggs	12.9	0	12.9	21.7	0	21.7
	Nuts, Beans & Soy	0.5	0.5	25.0	1.8	1.8	1.8
	Processed Meat	6.4	0	0	9.1	0	0
	Seafood/Meat	2.2	2.2	2.2	4.9	4.9	4.9
Snacks & Sweets	Candy	0.6	0.6	0.6	0.3	0.3	0.3
	Crackers	0.5	0.5	0	0.4	0.4	0
	Other Desserts	0.3	0.3	0.3	1.0	1.0	1.0
	Savory Snacks	0.9	0.9	0.9	0.7	0.7	0.7
	Snack/Meal Bars	0.4	0.4	0.4	1.0	1.0	1.0
	Sweet Bakery	20.6	2.9	0.6	12.8	2.5	2.9
Sugars	Sugars	7.5	7.5	7.5	8.6	8.6	8.6
Vegetables	Vegetables, Non-potato	0.4	0.4	0.4	2.2	2.2	2.2
	White Potatoes	1.7	1.7	1.7	8.7	8.7	8.7

**Table 3 nutrients-11-01374-t003:** Mean intake of nutrients at breakfast at T1 of the NRF 9.3 score and for optimized diets (absolute and relative model).

Nutrient	Children			Adults			
	T1	LP-R	LP-A	T1	LP-R	LP-A	Guidelines
Energy (kcal)	440.9	500.0	500.0	480.7	500.0	489.1	(300,500)
Added Sugar (g)	4.7	5.1	3.6	4.9	5.4	4.7	
Carbohydrates (g)	60.7	88.0	73.6	61.3	84.5	77.3	
PUFA (g)	3.3	1.9	3.7	4.2	2.4	2.9	
MUFA (g)	5.8	3.4	6.3	7.1	4.0	4.9	
Saturated Fat (g)	6.1	5.1	5.6	6.5	5.2	5.4	
Proteins (%)	12.4	12.8	15.0	13.9	13.1	14.2	
Carbohydrates (%E)	55.1	70.4	58.9	51.0	67.6	63.2	(55,75)
Added Sugars (%E)	4.3	4.1	2.9	4.1	4.3	3.8	<10
Total Fat (%E)	33.7	20.4	30.0	36.3	22.8	26.5	(20,30)
SFA (%E)	12.4	9.2	10.0	12.3	9.3	10.0	<10
Proteins (g)	13.7	16.0	18.7	16.7	16.4	17.4	>10
Dietary Fiber (g)	2.6	5.6	6.1	3.1	5.6	5.8	>5.6
Sodium (mg)	630.9	460.0	460.0	742.7	460.0	460.0	<460
Vitamin A (g)	195.3	373.9	284.9	186.1	316.9	360.3	>90
Thiamin (mg)	0.4	0.7	0.5	0.4	0.6	0.6	(>0.3,>0.2)
Riboflavin (mg)	0.6	1.1	0.9	0.7	1.1	1.2	(>0.5,>0.4)
Niacin (mg)	5.0	7.5	6.5	5.8	7.6	7.9	>4
Vitamin B6 (mg)	0.5	1.0	0.8	0.5	1.0	1.1	>0.3
Vitamin B12 (g)	1.5	3.2	2.4	1.4	3.0	3.3	(>0.6,>0.5)
Vitamin C (mg)	15.4	43.9	35.7	16.9	32.9	32.7	>18
Vitamin D (µg)	1.9	4.8	4.0	1.6	4.0	4.0	>4
Folate (g)	103.5	191.4	184.7	100.7	187.6	230.9	>80
Calcium (mg)	250.9	489.5	421.2	223.7	423.8	390.3	(>390,>325)
Iron (mg)	4.2	6.2	6.3	4.0	5.7	9.6	>3.6
Potassium (mg)	434.4	940.0	940.0	578.9	966.2	940.0	>940
Magnesium (mg)	45.7	84.0	98.6	61.9	96.1	95.7	>84
Zinc (mg)	2.2	4.2	3.8	2.4	3.8	4.7	>2.2

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
