# Peer review of "Designing Optimal Breakfast for the United States Using Linear Programming and the NHANES 2011–2014 Database: A Study from the International Breakfast Research Initiative (IBRI)"

_nutrients, 2019, doi:10.3390/nu11061374_

Reviewer 1 Report

This study by Vieux et al applied, in the large nationally representative National Health and Nutrition Examination Survey (NHANES), linear-programming (LP-) breakfasts with the original idea to evaluate whether such technique would resemble existing breakfast in the top tertile (T3) of diet quality, or whether LP-breakfasts would follow different pathways. What is more interesting about this paper is that the authors addressed the concept that following dietary guideline based on existing “healthy” eating habits would indeed be more feasible, and effective to implement than following the dietary recommendation that do not reflect the culinary origin/tradition of the studied population.

The MS is very well written, the study is correctly performed and data are well presented and interpreted.

However, I have some concerns regarding the discussion section which provides just a summary of the results. Essentially, the discussion lacks both the description and interpretation of the findings in light of what is known about the research topic. Besides, the authors did not provide an explanation of any new insights about the research topic after having taken into account the main findings. 

Author Response

Reviewer 1 :

This study by Vieux et al applied, in the large nationally representative National Health and Nutrition Examination Survey (NHANES), linear-programming (LP-) breakfasts with the original idea to evaluate whether such technique would resemble existing breakfast in the top tertile (T3) of diet quality, or whether LP-breakfasts would follow different pathways. What is more interesting about this paper is that the authors addressed the concept that following dietary guideline based on existing “healthy” eating habits would indeed be more feasible, and effective to implement than following the dietary recommendation that do not reflect the culinary origin/tradition of the studied population.

The MS is very well written, the study is correctly performed and data are well presented and interpreted.

However, I have some concerns regarding the discussion section which provides just a summary of the results. Essentially, the discussion lacks both the description and interpretation of the findings in light of what is known about the research topic. Besides, the authors did not provide an explanation of any new insights about the research topic after having taken into account the main findings. 

Answer: The discussion section was rewritten to take these points into account.  The opening of the Discussion section now reads

The present analyses showed that breakfasts associated with higher-quality diets were replicated, for the most part, through LP modeling.  The Nutrient Rich Food Index (NRF) was the measure of diet quality.  Lower-quality diets were those in the bottom tertile of NRF scores (T1), whereas higher-quality diets were those in the top tertile of NRF scores (T3).  Our approach was to use breakfasts associated with lower-quality diets (T1) as the point of departure and to compare breakfasts associated with higher-quality diets (T3) to those generated by two LP models.  As expected, breakfasts associated with T3 of NRF scores were associated with higher intakes of some key nutrients, including those that were in the NRF model and some that were not.  The T3 breakfasts also had more food groups of interest, notably fruit, dairy and whole grains.

Results showed that the LP models were able to improve the observed breakfast quality even more. Even though breakfast was already a relatively nutrient rich meal, the observed T3 breakfasts in children had below-recommended values for fiber, calcium, potassium, magnesium and vitamin D and had excess added sugars, as compared to IBRI recommendations.  For adults T3 breakfasts had too little potassium and vitamin D and too much sodium and added sugar, again as compared to IBRI recommendations.  The LP modeling showed that the observed breakfasts could be further improved, largely by changing the amounts of frequently eaten breakfast foods.

In past studies, LP models have been applied to the optimization of daily food patterns, subject to a variety of nutritional, economic and social constraints. LP models have also been used on the supply side, e.g. to optimize nutrient density of a school’s entire food supply  https://www.preprints.org/manuscript/201903.0178/v1  The present innovation was to apply LP modeling to optimize the quality of a single meal, breakfast, as opposed to optimizing the quality of the total diet.  Our use of two LP objective functions was meant to show that healthy breakfasts could be arrived at in multiple ways and following different food choices.  The second innovation was to base LP modeling on those breakfast foods that were already being consumed by children and adults in the NHANES database.

Reviewer 2 Report

Through use of linear programming models, this study identifies how poor quality breakfasts may be nutritionally improved (e.g. more likely to meet nutrient recommendations) in relation to their food compositions.  I commend the authors on taking a meals-based approach to examining diet quality, which has the potential to inform food-based dietary guidelines at the level of a meal. While the paper is generally well-conducted and of importance to its field, there are areas of the methods that require further detail and clarity.

 Introduction

1.      “In general, dietary guidance that is based on existing eating habits is more feasible and easier to implement than is dietary guidance that breaks entirely with habit, tradition, and culture”

I would like to see this comment further discussed in relation to the public health implications of the information generated from this study. How do you propose that the information could be used (practically) to improve population diet quality? Some context about how the existing dietary guidelines are framed and how taking a meals-based approach may be beneficial could be better explored. Doing this in the introduction and discussion sections of the paper would help clarify potential public health implication of the study’s findings and the need for future research in this space.

 Methods

2.      The study uses 31 food group categories that presented in Table 2 to describe the breakfast food compositions. However, a nutrient, rather than food-based, index is used as a measure of diet quality, based on 9 qualifying and 3 limiting nutrients. Given that dietary guidelines are food-based, the rationale for using a nutrient-based approach to determine overall diet quality and to optimise breakfast quality needs further clarification. How does the NRF9.3 compare with food-based diet quality indices such as the HEI - would participants be ranked similarly using both approaches?

3.      A number of abbreviations are used that are not fully clear/expanded at first use. These include NR, LIM, & D in the equations, and LPA-A and LPA-R in the figures

 Results

 4.      Saturated fat (SFA) is one of the limiting nutrients, yet food sources of SFAs are an important health consideration (e.g. SFA derived from dairy vs SFA derived from processed meats). The optimized diets result in lower SFA intakes, but it is not fully clear how this was achieved by looking at the food group data in the results. Intakes of processed meats decreased but milks increased substantially – both are sources of SFA and interpretation would be aided if milks were classified according to their fat content. Impact of fibre intakes, would also be enhanced by further categorising the grain foods that are listed in Table 2 into whole grain or refined grain. RTE breakfast cereals are classified according to sugar but not whole grain/refined grain, and again, this hinders the interpretation of the findings.

 5.      Interpreting figure 2 and Table 2 would be easier if the food groups labels used in the figure matched those in Table 2. For example, in figure 2 it is difficult to decipher which food categories contribute to the wholegrains category and which foods contribute to the refined grains category and if meat includes all types of meat. Another option could be a supplementary table that describes the main food sources contributing to the categories in Figures 1 & 2

 6.      High sugar RTE cereals sharply increased in the relative LP-optimized models but not the absolute models. Whilst these cereals may be a source of other nutrients, high sugar intakes are of concern and guidelines recommend that added sugars be limited. The LP-R model prioritizes foods already consumed (e.g. this suggests high sugar cereals are more prevalently consumed than low sugar ones) but the practical public health implications of basing recommendations for improving the nutritional quality of breakfast based on this approach may be problematic. That is promoting consumption of high sugar RTE cereals is incongruent with the message to limit added sugars. I suggest more in depth discussion of these issues is needed, particularly in penultimate paragraph of the discussion.

 Discussion

 7.      “By providing results using different mathematical functions, our study defends the possibility….in order to improve the nutritional quality of breakfast”.

This meaning of this sentence is unclear; please consider revising.

 The limitations are briefly discussed but could be better explained. For example, a self-report definition is cited as a limitation and it is implied that a time-of-day definition may be better, but the reasons are not clear. Similarly, it is unclear why food groups based on MyPlate food categories is a limitation.

 Author Response

Reviewer 2 :

Through use of linear programming models, this study identifies how poor quality breakfasts may be nutritionally improved (e.g. more likely to meet nutrient recommendations) in relation to their food compositions.  I commend the authors on taking a meals-based approach to examining diet quality, which has the potential to inform food-based dietary guidelines at the level of a meal. While the paper is generally well-conducted and of importance to its field, there are areas of the methods that require further detail and clarity.

 Introduction

1.      “In general, dietary guidance that is based on existing eating habits is more feasible and easier to implement than is dietary guidance that breaks entirely with habit, tradition, and culture”

 I would like to see this comment further discussed in relation to the public health implications of the information generated from this study. How do you propose that the information could be used (practically) to improve population diet quality? Some context about how the existing dietary guidelines are framed and how taking a meals-based approach may be beneficial could be better explored. Doing this in the introduction and discussion sections of the paper would help clarify potential public health implication of the study’s findings and the need for future research in this space.

 Answer: Thank you for that comment. We now say

 It has often been a concern that dietary recommendations, issued by expert bodies or national governments are hard to follow and may not actually feasible from nutritional, behavioral or economic standpoints.  First, there were cases where multiple nutrient requirements could not be satisfied simultaneously (ref).  Second, healthy foods were sometimes proposed in unrealistic large amounts (ref).  Another concern was that the recommended food patterns were simply too expensive (ref).  One advantage of LP models is their ability to reconcile these multiple demands without departing too far from existing eating habits.  In the present study, we were able to show that the IBRI breakfast recommendations were feasible in the sense that a mathematical solution was available.  In other words, we validated a set of nutrient based recommendation by selecting the right combination of breakfast foods, with emphasis on those foods that were already consumed by children and adults.   Food based advice on healthy breakfasts can have practical implications. Healthy breakfast foods can be the starting point of public health politics. For example, in France, a free breakfast will be provided to a sample of children in different school. The results from the present study may help to design the breakfast that we be provided to children.

 Methods

2.      The study uses 31 food group categories that presented in Table 2 to describe the breakfast food compositions. However, a nutrient, rather than food-based, index is used as a measure of diet quality, based on 9 qualifying and 3 limiting nutrients. Given that dietary guidelines are food-based, the rationale for using a nutrient-based approach to determine overall diet quality and to optimise breakfast quality needs further clarification. How does the NRF9.3 compare with food-based diet quality indices such as the HEI - would participants be ranked similarly using both approaches?

Answer: This is a very interesting and timely comment.  We say

 The present food based approach aligns with the current dietary recommendations and guidelines which have become more food- as opposed to nutrient-based.  The current research emphasis is on likewise on food patterns as opposed to individual nutrients.  Even though nutrient profiling models continue to be wholly nutrient based, ca case can be made (Drewnowski et al. Nutrition Reviews 77:404-416 June 2019) for advancing a hybrid NP approach that takes both nutrients and desirable food groups into account.  Hybrid nutrient density scores will provide for a better alignment between NP models and the DGA, a chief instrument of food and nutrition policy in the United States. Such synergy may lead ultimately to improved dietary guidance, sound nutrition policy, and better public health.” 

 So the NP approach is evolving very much along the lines indicated by the Reviewer.  In a previous paper in the IBRI series we did analyze the correlation between NRF and HEI 2015 score and its subscores.  The correlation was in the order of 0.43 .and was statistically significant..

3.      A number of abbreviations are used that are not fully clear/expanded at first use. These include NR, LIM, & D in the equations, and LPA-A and LPA-R in the figures

 Answer: NR stands for“Nutrientsto encourage”, LIM stands for “Nutrients to limit”, D stands for “Distance”, LP-Astands for “Linear programming model with an absolute objective function”, LP-R stands for “Linear programming model with a relative objective function”

 Results

 4.      Saturated fat (SFA) is one of the limiting nutrients, yet food sources of SFAs are an important health consideration (e.g. SFA derived from dairy vs SFA derived from processed meats). The optimized diets result in lower SFA intakes, but it is not fully clear how this was achieved by looking at the food group data in the results. Intakes of processed meats decreased but milks increased substantially – both are sources of SFA and interpretation would be aided if milks were classified according to their fat content. Impact of fibre intakes, would also be enhanced by further categorising the grain foods that are listed in Table 2 into whole grain or refined grain. RTE breakfast cereals are classified according to sugar but not whole grain/refined grain, and again, this hinders the interpretation of the findings.

Answer: This is a very good point.  The US Department of Agriculture has different ways of assigning foods into aggregate categories.  We chose to use the What We Eat in America (WWEIA) scheme, since it was more granular that the USDA MyPyramid scheme and therefore better suited to linear programming.  Whereas the MyPyramid scheme does split grains into refined and whole grains (see out paper in Nutrients 2019), the present WWEIA scheme does not. By contrast, the present scheme separates dairy and protein foods into multiple categories, whereas MyPyramid does not.  In the present LP model, the frequency weighted “milk” contained about 1g of SFA and 48kcal for 100g. In order to reduce SFA intake, milk  effectively replaced cheese (which contains about 12g of SFA and 290kcal per 100g and which was removed in absolute function models), as well as processed meat (7g of SFA and 290kcal per 100g and which was removed in all model).  MIlk also replaced sweet bakery (7g of SFA and 400kcal per 100g), eggs (4.5g of SFA and 180kcal per 100g) and mixed dishes (3.7g of SFA and about 215kcal per 100g).

 The USDA scheme of separating RTE cereals by sugar content did capture whole grains. On average, 100g of RTEC high in sugars contains 1.21 once of whole grain and 0.82 once of refined ones and 100g of RTEC low in sugars contains 1.94 once of whole grain and 0.91 once of refined ones

5.      Interpreting figure 2 and Table 2 would be easier if the food groups labels used in the figure matched those in Table 2. For example, in figure 2 it is difficult to decipher which food categories contribute to the wholegrains category and which foods contribute to the refined grains category and if meat includes all types of meat. Another option could be a supplementary table that describes the main food sources contributing to the categories in Figures 1 & 2.

Answer: Following reviewer suggestion, 6 supplemental tables were added. The tables display the contribution of WWEIA categories to MyPyramid food groups in observed and both optimized diets in children and adults. They show that contribution of processed meat to meat MyPyramid food groups is substituted with Meat and seafood WWEIA category.

 6.      High sugar RTE cereals sharply increased in the relative LP-optimized models but not the absolute models. Whilst these cereals may be a source of other nutrients, high sugar intakes are of concern and guidelines recommend that added sugars be limited. The LP-R model prioritizes foods already consumed (e.g. this suggests high sugar cereals are more prevalently consumed than low sugar ones) but the practical public health implications of basing recommendations for improving the nutritional quality of breakfast based on this approach may be problematic. That is promoting consumption of high sugar RTE cereals is incongruent with the message to limit added sugars. I suggest more in depth discussion of these issues is needed, particularly in penultimate paragraph of the discussion.

 Answer:  That is a good point – and we agree.  But the problem lies with the USDA terminology and the split in the WWEIA database (see:

https://www.ars.usda.gov/ARSUserFiles/80400530/pdf/1314/food_category_list.pdf.  We now make the specific point that the “high sugar” cereals actually contain more whole grains and there is no intent here to promoted added sugar – rather the contrary since added sugar was in fact minimized by the LP models.

 Discussion

 7.      “By providing results using different mathematical functions, our study defends the possibility….in order to improve the nutritional quality of breakfast”.

This meaning of this sentence is unclear; please consider revising.

 The limitations are briefly discussed but could be better explained. For example, a self-report definition is cited as a limitation and it is implied that a time-of-day definition may be better, but the reasons are not clear. Similarly, it is unclear why food groups based on MyPlate food categories is a limitation.

 Answer: In past research we have used both lean definition and the time of the eating occasion.  The two do overlap.  For this study, based on the breakfast meal, we elected to go with meals.  Self report is always a weakness in dietary assessment studies, no matter what.  We also elected to use the richer WWEIA food groups as opposed to the narrower MyPyramid aggregation schemes in order to provide the LP model with more food-based breakfast options that could be used for dietary  guidance.

Round  2

Reviewer 1 Report

The authors addressed the main concerns from the reviews